# The association between tropical cyclones and dengue fever in the Pearl River Delta, China during 2013-2018: A time-stratified case-crossover study

**Chuanxi Li**[1,2], **Qi Zhao**[1,2], **Zhe Zhao**[1,2], **Qiyong Liu**[2,3], **Wei Ma**[1,2]*

**1** Department of Epidemiology, School of Public Health, Cheeloo College of Medicine, Shandong University, Jinan, China, **2** Shandong University Climate Change and Health Center, Jinan, China, **3** State Key Laboratory of Infectious Disease Prevention and Control, National Institute for Communicable Disease Control and Prevention, Chinese Center for Disease Control and Prevention, Beijing, China

* weima@sdu.edu.cn

**Data Availability Statement:** The dengue fever data underlying the results presented in the study cannot be shared publicly because of the limitation of data availability in the data management rule of

## Abstract

### Background

Studies have shown that tropical cyclones are associated with several infectious diseases, while very few evidence has demonstrated the relationship between tropical cyclones and dengue fever. This study aimed to examine the potential impact of tropical cyclones on dengue fever incidence in the Pearl River Delta, China.

### Methods

Data on daily dengue fever incidence, occurrence of tropical cyclones and meteorological factors were collected between June and October, 2013–2018 from nine cities in the Pearl River Delta. Multicollinearity of meteorological variables was examined via Spearman correlation, variables with strong correlation ($r$>0.7) were not included in the model simultaneously. A time-stratified case-crossover design combined with conditional Poisson regression model was performed to evaluate the association between tropical cyclones and dengue fever incidence. Stratified analyses were performed by intensity grades of tropical cyclones (tropical storm and typhoon), sex (male and female) and age-groups (<18, 18–59, $\geq$60 years).

### Results

During the study period, 20 tropical cyclones occurred and 47,784 dengue fever cases were reported. Tropical cyclones were associated with an increased risk of dengue fever in the Pearl River Delta region, with the largest relative risk of 1.62 with the 95% confidence interval (1.45–1.80) occurring on the lag 5 day. The strength of association was greater and lasted longer for typhoon than for tropical storm. There was no difference in effect estimates between males and females. However, individuals aged over 60 years were more vulnerable than others.

Chinese Center for Disease Control and Prevention. Access to these data may be requested through the Chinese Center for Disease Control and Prevention for researchers who meet the criteria for access to confidential data. Researchers who would like to access the data can contact the E-mail: liuqiyong@icdc.cn. The tropical cyclones data are available from the China Weather Typhoon Network (http://typhoon.weather.com.cn/). The meteorological data are available from the China Meteorological Data Service Center (http://data.cma.cn/).

**Funding:** This study was supported by the National Natural Science Foundation of China (grant number 82073615) to WM and the State Key Laboratory of Infectious Disease Prevention and Control (grant number 2018SKLID302) to QL. The funders had no role in study design, data collection and analysis, decision to publish, or preparation of the manuscript.

**Competing interests:** The authors have declared that no competing interests exist.

## Conclusions

Tropical cyclones are associated with increased risk of local dengue fever incidence in south China, with the elderly more vulnerable than other population subgroups. Health protective strategies should be developed to reduce the potential risk of dengue epidemic after tropical cyclones.

## Author summary

Dengue fever, a mosquito-borne tropical infectious disease, has been increasingly serious in recent decades, causing great healthcare burden in low-latitude regions and countries. *Aedes* is the vector of dengue fever, particularly sensitive to climatic conditions during all stages of the life cycle. Numerous epidemiological studies have demonstrated the association between dengue fever and meteorological factors (e.g., temperature, precipitation and relative humidity). Tropical cyclones are a common extreme weather events in the low latitude and have been associated with the outbreak of several infectious diseases. However, the impact of tropical cyclones on the incidence of dengue fever has not been well clarified. In this study, we explored the association between tropical cyclones and dengue fever in the Pearl River Delta region, China. The results showed that the local incidence of dengue fever was substantially associated with tropical cyclones over a certain lag period, with the effect estimate greater for stronger tropical cyclones. The elderly was more vulnerable than any other population subgroups. The findings highlighted the importance of developing public health surveillance, preparedness, and response targeting the outbreak of dengue fever during the tropical cyclone season.

## Introduction

Tropical cyclones are one of the most destructive extreme weather events, threatening population health in tropical and subtropical coastal areas [1]. China is located in the east of the Eurasian continent and borders on the northwestern Pacific Ocean. The special location and long coastline make it one of the countries most affected by tropical cyclones in the world: In 2018 alone, over 32 million Chinese were affected by tropical cyclones, leading to a direct economic loss of $11 billion [2]. In China, Guangdong Province experiences more loss than any other provinces since half of tropical cyclones land in this place [3]. In addition to the direct threat to life, tropical cyclones may also affect the outbreak of certain infectious diseases, e.g., acute hemorrhagic conjunctivitis, infectious diarrhea, leptospirosis, and hand, foot and mouth disease [4–7]. For instance, a study has found that tropical storms may increase the risk of hand, foot and mouth disease among children under six years old in Guangdong Province, China [8]. Another study in Guangdong Province has demonstrated that all grades of tropical cyclones could boost the risk of infectious diarrhea [9].

Dengue fever (DF) is an acute mosquito-borne viral disease transmitted via *Aedes aegypti* or *Aedes albopictus* [10]. Symptoms included high fever, severe headache, muscle and joint pains, with severe cases being hospitalized or even fatal. In recent decades, the incidence of DF has grown dramatically worldwide. The World Health Organization (WHO) has listed DF among the top ten diseases in 2019 [11]. As a tropical infectious disease, DF is causing an increasing disease burden in China, especially in the southeast coastal region. In 2014,

Guangdong Province experienced an unprecedented epidemic of DF, resulting in more than 40,000 infections [12]. In addition, about 94% of the local cases in China during 2007–2017 were reported in Guangdong Province, and more than 90% of those cases were from the Pearl River Delta region (PRD)–the largest urban agglomeration in area and population of the world [13–15].

So far, numerous studies have shown that DF and its biological vector are both closely related to meteorological factors and climate change [16–18]. From 1950 to 2018, the global climate suitability for the transmission of DF increased by 8.9% for *Aedes aegypti* and 15.0% for *Aedes albopictus* [19]. However, very few studies have evaluated the impact of tropical cyclones on the incidence of DF. The transmission of DF is related to tropical cyclones via several potential pathways. The moderate to heavy rainfalls following tropical cyclones may increase mosquito breeding sites. Meanwhile, the destruction of tropical cyclones may increase the population's displacement, their exposure risk to mosquito bites and delay in access to healthcare services. For example, the hurricane Michelle may potentially facilitate the indigenous spread of the dengue serotype DENV-3 in Havana [20]. Between 1951 and 2017, 490 tropical cyclones landed in China, equating to 7.4 times per year on average [21]. Provided the frequency and intensity of tropical cyclones are expected to increase under the context of climate change [22], it is necessary to quantify the effect size of the impact of tropical cyclones on DF-related burden in PRD.

In this study, we examined the relationship between tropical cyclones and DF incidence in the PRD using a time-stratified case-crossover design with a conditional Poisson regression model. The case-crossover design has been widely used by epidemiological studies to explore the association between various meteorological risks and health outcomes (e.g., vector-borne diseases) [23–25]. It is especially suitable for exploring the effect of short-term exposure on acute events. The principle is that each patient is treated as a stratum, where the exposure during the risk period is compared with exposures during the control periods [26]. Meanwhile, we investigated the modifying effects of sex, age and tropical cyclone intensity on the association.

## Materials and methods

### Ethical statement

The unidentifiable data on DF surveillance were obtained from the National Notifiable Disease Surveillance System (NDSS), the Chinese Center for Disease Control and Prevention. This study was approved by the Ethical Review Committee of School of Public Health in Shandong University (LL20200410).

### Study area and study period

The PRD is a low-lying area surrounding the Pearl River estuary, located in the south central of Guangdong Province and adjacent to Hong Kong and the Macao Special Administrative Region (Fig 1). It contains nine cities–Guangzhou, Shenzhen, Zhuhai, Foshan, Jiangmen, Zhaoqing, Huizhou, Dongguan, and Zhongshan. By the end of 2018, over 63 million residents lived in the PRD, accounting for 55.5% of the whole population in Guangdong [27]. The PRD has a subtropical monsoon climate, characterized by hot and humid summer while warm and dry winter. The PRD is the key transportation hub between the South Mainland China and other tropical/subtropical countries, resulting in a large number of DF cases imported from overseas. In this study, we chose June to October as the study period considering all tropical cyclones landed during this period and the meantime highest DF incidence in the PRD of the year.

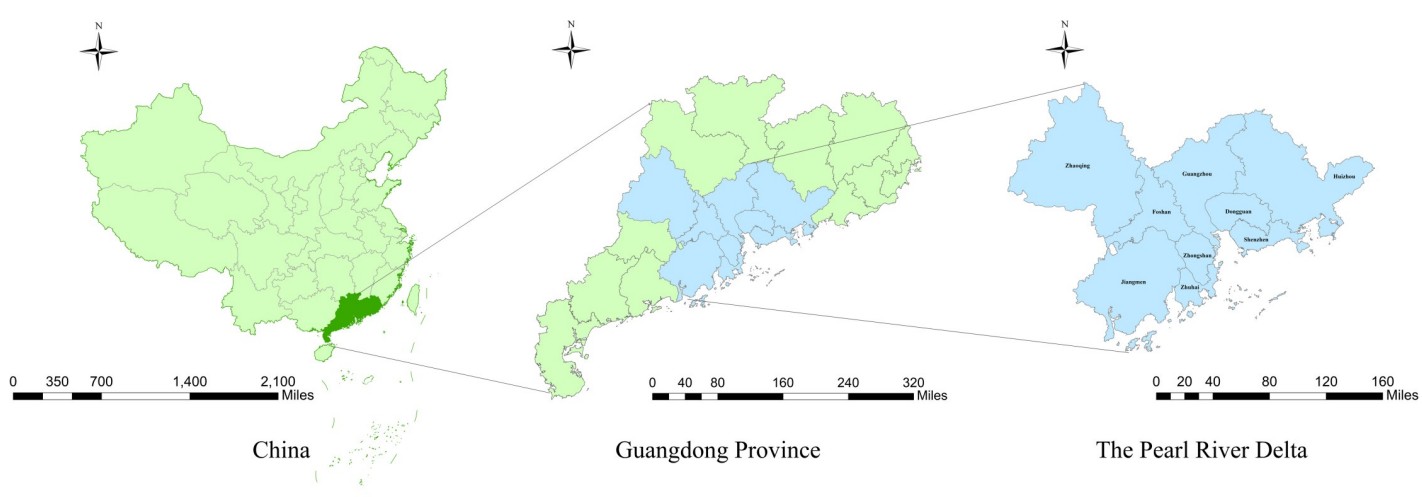

**Fig 1. Location of the study area in China.** The base layer of the map is from Resource and Environment Science and Data Center (http://www.resdc.cn/data.aspx?DATAID=201).

### Disease data

Data on DF infections during June to October, 2013–2018 in the PRD were obtained from NDSS (coded as A90 in the Tenth Revision of the International Classification of Diseases), including information on sex, age, date of onset, location of disease occurrence and epidemiological history. By the Chinese legislation, DF is a notifiable category B infectious disease, and all cases of DF must be reported to the local Center for Disease Control and Prevention within 24 hours [28]. All DF cases included in the study were diagnosed according to the diagnostic criteria issued by the National Health Commission. In this study, we excluded the imported DF cases, i.e., cases going abroad or leaving the PRD region within 14 days before the day of onset. Their infections onsets were considered unrelated to the local environmental context.

### Tropical cyclones and meteorological data

Given the special location and land size of PRD in Guangdong Province and the tracks of these cyclones, all tropical cyclones landing in Guangdong Province were considered to affect the study area and thus selected for analysis in this study. Following previous studies, we used the Beaufort scale to define the exposure period of tropical cyclones [29], i.e., the time interval from the date when the wind force circle at Beaufort level 7 (near gale-force wind speed; $\geq$28 knots) arrived the PRD region to the date when tropical cyclones left or disappeared. Basic information of tropical cyclones was collected from the Yearbook of Tropical Cyclone and the China Weather Typhoon Network (http://typhoon.weather.com.cn/) maintained by the China Meteorological Administration. The China Meteorological Administration classified tropical cyclone into six strength levels according to the maximum wind velocity near the bottom of tropical cyclone center, namely tropical depression (10.8–17.1 m/s), tropical storm (17.2–24.4 m/s), severe tropical storm (24.5–32.6 m/s), typhoon (32.7–41.4 m/s), severe typhoon (41.5–50.9 m/s), and super typhoon ($\geq$51 m/s) [30]. During the study period there were no tropical cyclones with strength belonging to tropical depression at the landing time. To improve the modeling stability, tropical cyclones belonging to tropical storm and strong tropical storm at the time of landing were combined into the tropical storm category, and tropical cyclones belonging to typhoon, severe typhoon and super typhoon were classified into the typhoon category.

Daily monitoring meteorological data during the same period were collected from the China Meteorological Data Sharing Service System (http://cdc.cma.gov.cn/). We obtained 13 daily meteorological variables from 11 local ground weather stations, including minimum, average and maximum temperature (MiT, AT and MaT, ˚C), average precipitation (RF, mm), average, maximum and extreme wind velocity (AWV, MaWV and EWV, m/s), minimum, average and maximum air pressure (MiAP, AAP and MaAP, hPa), average vapor pressure (AVP, hPa), minimum and average relative humidity (MiRH and ARH, %). The daily mean values of all meteorological factors of 11 weather stations were then calculated to represent the general climatic conditions in the PRD region.

## Statistical analysis

To minimize multicollinearity, Spearman correlation analysis was performed to evaluate the correlation among tropical cyclones and meteorological variables. Variables with strong correlation ($r>0.7$) were not included in the final model simultaneously.

A time-stratified case-crossover design combined with conditional Poisson regression model was used to evaluate the association between tropical cyclones and DF incidence, with effect estimates described using risk ratios (*RRs*) with 95% confidence intervals (*CIs*). Case-crossover design reduces the selection bias through case self-matching and improves the efficiency of research [31,32]. In design, the control periods were defined as the same days of the week in the same calendar month of the onset date of DF, so that there were 3–4 controls before or after the onset day in each matched stratum. This bidirectional selection of controls further adjusted for long-term trend, seasonal variation and "weekday effect" [26]. Although conditional logistic regression is the most commonly used statistical method in case-crossover study, conditional Poisson regression has been proved to be a flexible alternative. The later can effectively adjust for the autocorrelation and overdispersion of time series data, reduce the size of parameters to be estimated without undermining the accuracy of effect estimates [33]. The regression model was as follows:

$$\log(Y_t) = \beta_0 + \beta_1 TC + \beta_2 AT + \beta_3 RF + \beta_4 AWV + \beta_5 EWV + \beta_6 AAP$$

where $Y_t$ denotes the daily number of DF cases on day *t*. *TC* is a categorical variable denoting different grades of tropical cyclones, $\beta_0$ is the intercept, $\beta_1, \beta_2, \ldots \beta_6$ are partial regression coefficients. In the study area, the average incubation period of DF was 5 days [34]. As with previous study [35], we used the lag range of 5 to 14 days to capture the lag structure of the effect estimates of tropical cyclones since the association over lag 0–4 days was less biologically plausible. Stratified analyses were performed by sex, age-groups (<18, 18–59, ≥60 years) and tropical cyclone intensity (tropical storm and typhoon).

Sensitivity analyses were conducted to evaluate the robustness of the model. In addition to Spearman correlation analysis, principal component analysis (PCA) was performed to eliminate the multicollinearity among meteorological factors. The principles of extracting principal components were: (1) the eigen value ≥1 and (2) the cumulative proportion reaching 80%. In addition, different combinations of meteorological variables were also used in the sensitivity analyses. Specifically, two alternative models were fitted by excluding RF and EWV on the basis of the original model, respectively.

All statistical analyses were conducted using R software (version 4.0.4) with the "gnm", "tsModel", "Epi" and "psych" packages. *P* values <0.05 (two-sided) were considered statistically significant.

**Table 1. Basic information of tropical cyclones affecting the PRD from 2013–2018.**

| Name | Grade | Landing site | Exposure Period |
|---|---|---|---|
| Rumbia | Severe Tropical Storm | Zhanjiang | $1^{st}$ - $2^{nd}$ Jul 2013 |
| Utor | Severe Typhoon | Yangjiang | $13^{th}$ - $16^{th}$ Aug 2013 |
| Usagi | Severe Typhoon | Shanwei | $22^{nd}$ - $23^{rd}$ Sep 2013 |
| Hagibis | Tropical Storm | Shantou | $15^{th}$ Jun 2014 |
| Rammasun | Super Typhoon | Zhanjiang | $18^{th}$ Jul 2014 |
| Kalmaegi | Severe Typhoon | Zhanjiang | $16^{th}$ Sep 2014 |
| Linfa | Typhoon | Shanwei | $9^{th}$ - $10^{th}$ Jul 2015 |
| Mujigae | Super Typhoon | Zhanjiang | $3^{rd}$ - $4^{th}$ Oct 2015 |
| Nida | Typhoon | Shenzhen | $1^{st}$ - $2^{nd}$ Aug 2016 |
| Dianmu | Tropical Storm | Zhanjiang | $18^{th}$ Aug 2016 |
| Haima | Typhoon | Shanwei | $21^{st}$ - $22^{nd}$ Oct 2016 |
| Merbok | Severe Tropical Storm | Shenzhen | $12^{th}$ - $13^{th}$ Jun 2017 |
| Hato | Severe Typhoon | Zhuhai | $23^{rd}$ - $24^{th}$ Aug 2017 |
| Pakhar | Severe Tropical Storm | Zhuhai | $27^{th}$ Aug 2017 |
| Mawar | Tropical Storm | Shanwei | $4^{th}$ Sep 2017 |
| Khanun | Severe Tropical Storm | Zhanjiang | $15^{th}$ Oct 2017 |
| Ewiniar | Tropical Storm | Yangjiang | $7^{th}$ - $9^{th}$ Jun 2018 |
| Bebinca | Tropical Storm | Zhanjiang | $12^{th}$ - $15^{th}$ Aug 2018 |
| Barijat | Severe Typhoon | Zhanjiang | $12^{th}$ - $13^{th}$ Sep 2018 |
| Mangkhut | Tropical Storm | Jiangmen | $16^{th}$ - $17^{th}$ Sep 2018 |

## Results

### Basic information of tropical cyclones and DF cases

During the study period, there were 20 tropical cyclones affecting the PRD region, including 6 tropical storms, 4 severe tropical storms, 3 typhoons, 5 severe typhoons and 2 super typhoons. Details of each tropical cyclones are displayed in Table 1.

There were 47,784 DF cases in the PRD during the study period. Table 2 shows the characteristics of the distribution of DF cases, with 50.4% being females and 78.9% aged between 18 and 59 years old. The incidence of DF in the PRD region had obvious interannual variation, which peaked in 2014 (n = 42,109) and bottomed in the following 2015 (n = 96) and 2016 (n = 225). The information of meteorological factors and the number of DF cases during each tropical cyclone is displayed in S1 Fig.

**Table 2. The number of DF cases during study periods in the PRD from 2013 to 2018.**

| | 2013 | 2014 | 2015 | 2016 | 2017 | 2018 |
|---|---|---|---|---|---|---|
| Sex | | | | | | |
| Male | 1,164 | 20,851 | 47 | 109 | 483 | 1,027 |
| Female | 1,301 | 21,258 | 49 | 116 | 491 | 888 |
| Age (years) | | | | | | |
| <18 | 227 | 3,921 | 1 | 12 | 84 | 173 |
| 18–59 | 1,881 | 30,669 | 83 | 168 | 701 | 1,430 |
| ≥60 | 357 | 7,519 | 12 | 45 | 189 | 312 |
| Total | 2,465 | 42,109 | 96 | 225 | 974 | 1,915 |

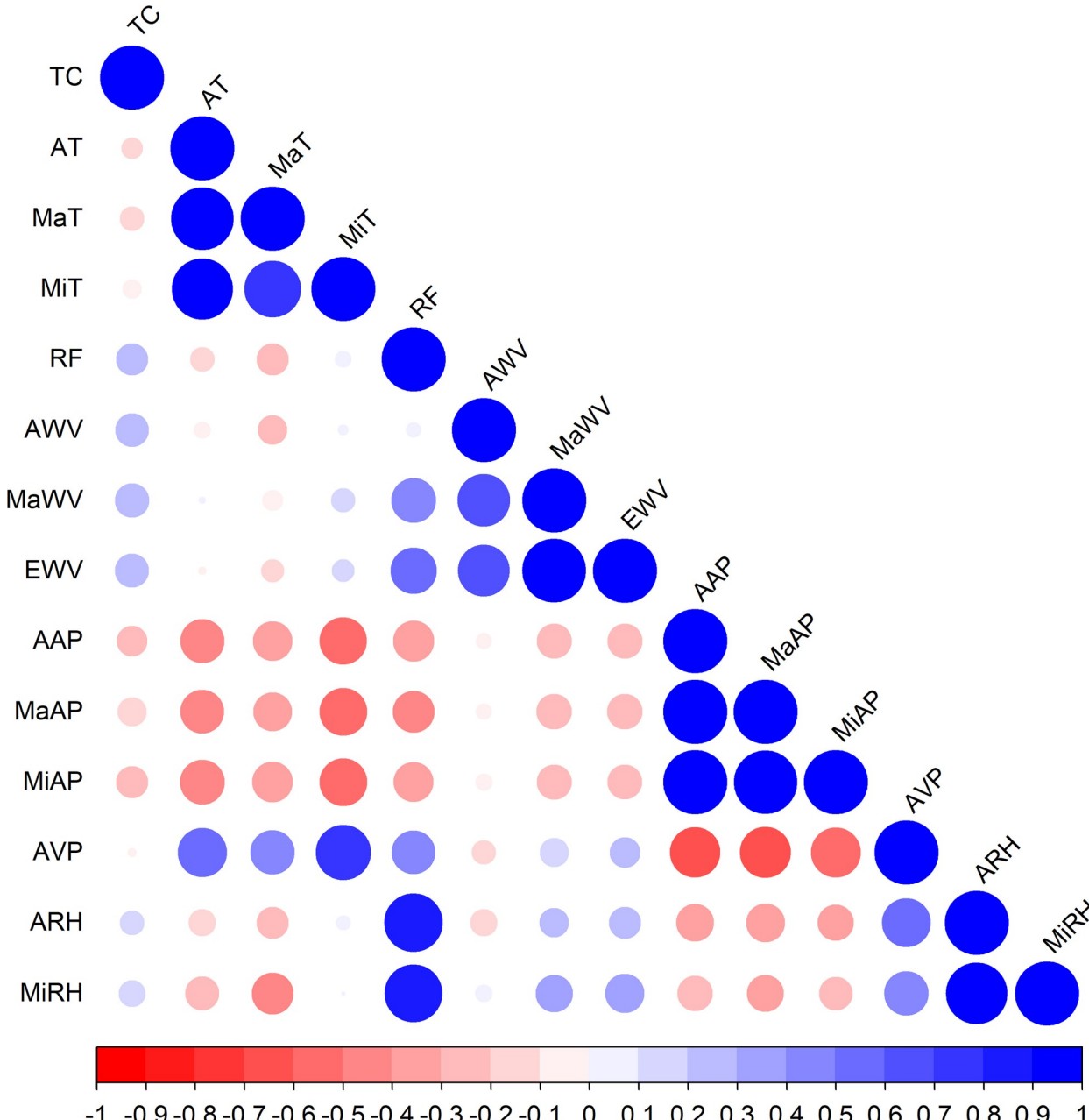

**Fig 2. Correlation of tropical cyclone intensities and meteorological variables.** TC: tropical cyclone intensities; AT: average temperature; MaT: maximum temperature; MiT: minimum temperature; RF: average precipitation; AWV: average wind velocity; MaWV: maximum wind velocity; EWV: extreme wind velocity; AAP: average air pressure; MaAP: maximum air pressure; MiAP: minimum air pressure; AVP: average vapor pressure; ARH: average relative humidity; MiRH: minimum relative humidity.

### Correlation analysis

Fig 2 shows the result of Spearman correlation test across meteorological factors. There was no strong correlation between TC and other factors ($r<0.3$). Following the previous literature, we retained meteorological covariates with greater biological rationality in the pair of variables with strong correlation ($r>0.7$). Finally, the variables included in the conditional Poisson

regression model were daily AT, RF, AWV, EWV, and AAP. The detailed Spearman correlation coefficients are shown in S1 Table.

## Effects of tropical cyclones on dengue fever

Fig 3 shows the *RR*s of tropical cyclones on DF incidence on lag 5–14 days. The results indicated that tropical cyclones were associated with increased risk of DF up to lag 9 day, with the largest effect estimate occurring on lag 5 day (*RR* = 1.62, 95%*CI*: 1.45–1.80) and lag 6 day (*RR* = 1.55, 95%*CI*: 1.40–1.72). Significant association between tropical storm and DF was only observed on lag 5 day (*RR* = 1.29, 95%*CI*: 1.04–1.59). The effect estimate of typhoon on DF was greater and lasted longer than tropical storm, with the strongest effect occurring on lag 6 day (*RR* = 1.71, 95%*CI*: 1.54–1.91).

Results of stratified analysis by sex and intensity grades of tropical cyclones are shown in Fig 4A and 4B. There was no significant relationship between tropical storms and DF for males and females. For typhoons, there was no evident difference in effect estimates by sex, with the strongest effects occurring on lag 6 day for both males (*RR* = 1.78, 95%*CI*: 1.53–2.07) and females (*RR* = 1.64, 95%*CI*: 1.41–1.91). Fig 4C and 4D display the results of subgroup analysis by age. Similarly, the impact of tropical storms on the incidence of DF was insignificant across all age-groups. However, the impact of typhoons on DF incidence varied. The strongest effect estimate was shown on lag 7 day (*RR* = 1.60, 95%*CI*: 1.13–2.28) for younger people under 18 years, and on lag 5 day (*RR* = 1.84, 95%*CI*: 1.61–2.09) for adults between 18–59 years. In comparison, people aged over 60 years was more vulnerable, with the highest effect estimate occurring on lag 6 day (*RR* = 2.05, 95%*CI*: 1.59–2.64).

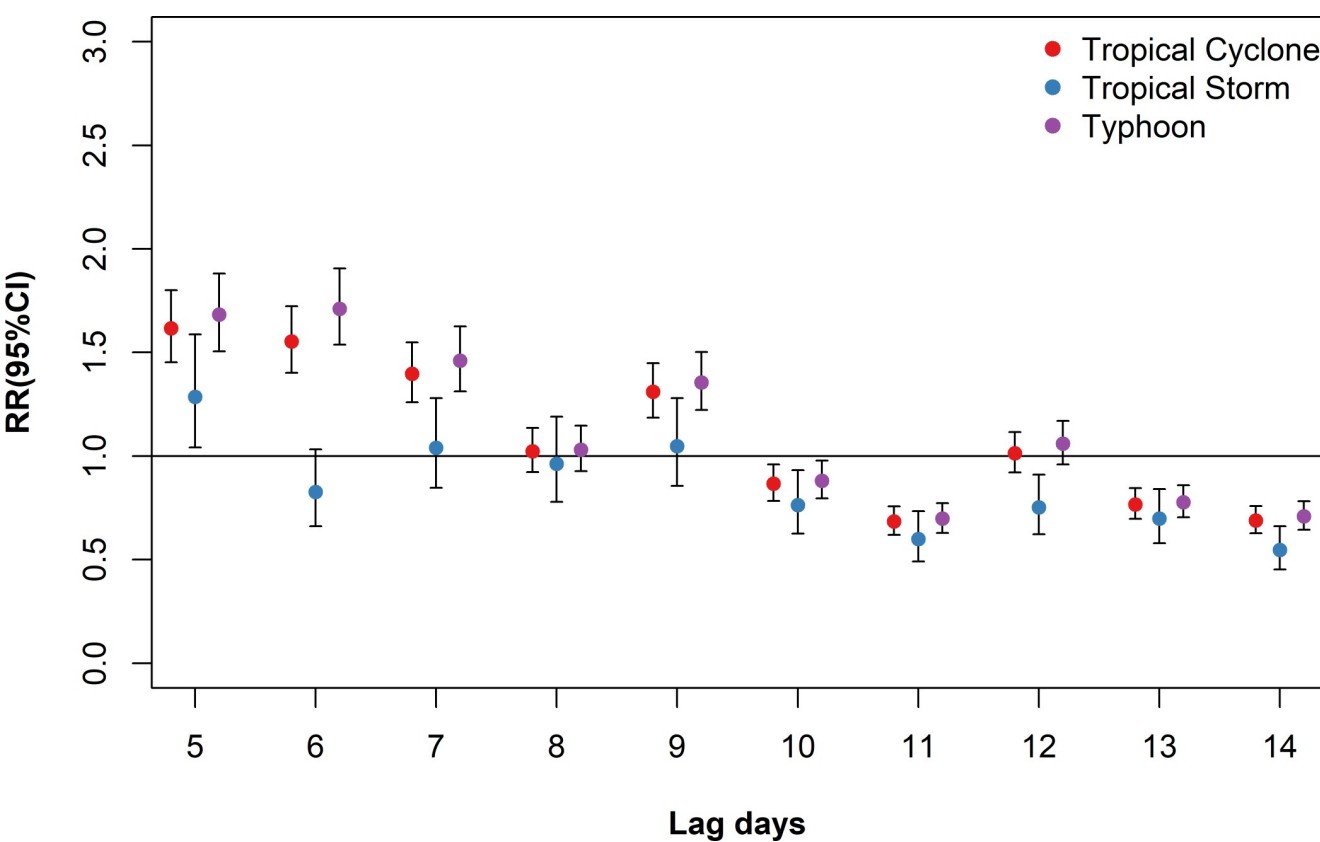

**Fig 3. Association between tropical cyclones and risk of DF incidence by intensity grades on lag 5–14 days in the PRD during the study period.**

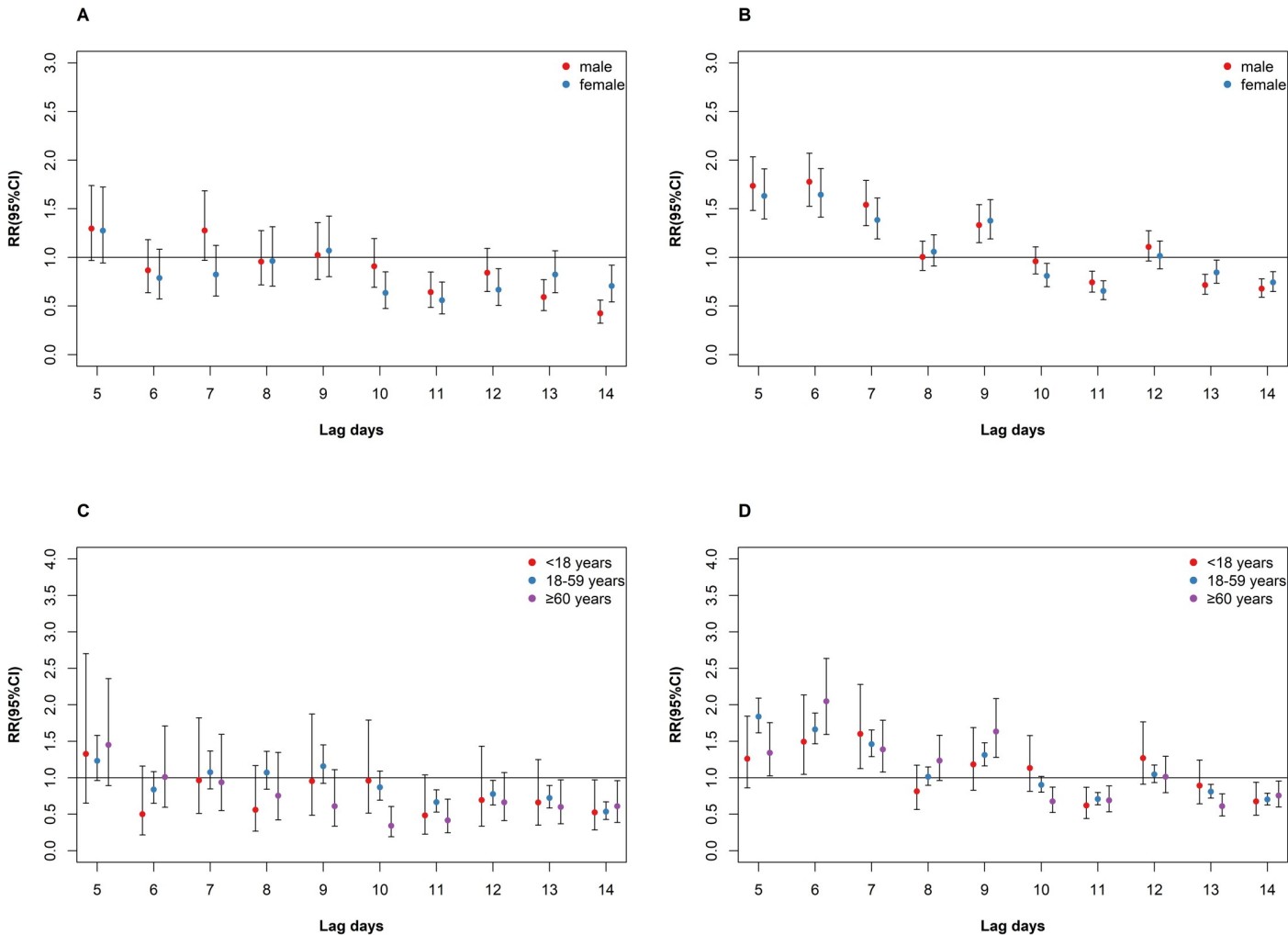

**Fig 4. Association between tropical cyclones and risk of DF incidence on lag 5–14 days in the PRD during the study period among population subgroups.** A: impact of tropical storms on DF by sex; B: impact of typhoons on DF by sex; C: impact of tropical storms on DF by age-groups; D: impact of typhoons on DF by age-groups.

## Sensitivity analysis

First, we fitted an alternative conditional Poisson regression model, including three principal components (PCs) according to the inclusion criteria to replace all meteorological variables (S2 Table). The effect estimates of tropical cyclones on DF on all lag days were similar to the original ones (S2 Fig). In addition, the results showed that there was little difference after excluding RF and EWV separately from the original model (S3 Fig). All of these results suggested the robustness of our findings.

## Discussion

In this study, we explored the relationship between tropical cyclones and DF incidence in the PRD, China. Our results suggested that both tropical storms and typhoons were associated with increased risk of DF incidence, with the greatest effect occurring on the lag 5 day. The impact of typhoons was stronger and lasted longer than tropical storms. There was no

substantial difference in effect estimates between males and females, while the elderly was more vulnerable than other age subgroups.

Our finding of the increased risk of DF incidence after the landing of tropical cyclone is in line with previous studies [7,36]. For instance, a study in Southeast China documented that tropical cyclone was likely to increase the risk of DF [7]. Another study indicated that tropical cyclones substantially contributed to the onset and magnitude of the unprecedented DF outbreak in Réunion in the south of Indian Ocean in 2018 [36]. As a mosquito-borne disease, the impact of tropical cyclones on DF incidence may reflect the combined vulnerability of vector and human beings. On one hand, tropical cyclones and the associated meteorological conditions are conducive to the survival and reproduction of mosquitoes. An experimental research in Puerto Rico found that the density of mosquitoes sharply increased in the next a few weeks after the landing of hurricane Maria [37]. However, the lack of necessary information at the vector level precludes our further deduction of the underlying mechanisms that may mediate this relationship. A preliminary study found that the number of *Aedes albopictus* positive breeding sites increased significantly after heavy precipitation [38], suggesting that the cyclone-related rainfall may be associated with the increase of mosquito density. On the other hand, storm surge may cause severe destruction to the houses, infrastructures, and anti-mosquito equipment, which further increase the risk of human-mosquito contact. A US study documented that 87% of households experienced more frequent mosquito biting after storms [39]. In addition, the threats of life safety, psychological trauma and economic losses associated with tropical cyclones are speculated to work comprehensively to reduce the awareness of affected households from the protectiveness of mosquito-borne diseases [39]. Future research should be performed to explore the effects of tropical cyclones and associated meteorological factors on the susceptibility of vectors and hosts for a systematic explanation of the complex triadic relationship.

In this study, we found that the effect estimate of typhoons on DF incidence was greater and lasted longer than tropical storms. Considering the intensity of tropical cyclones was based on the maximum wind velocity, this effect variation may indirectly reflect a positive correlation between wind speed and DF. Yet, this conclusion should be put forward cautiously due to the inconsistency with previous studies [40–43]. A national study in Thailand demonstrated that higher wind power may increase dengue incidence [40]. It's possible that strong wind on the sea or river surface increases evaporation area, and higher humidity would help mosquitoes survive longer and thus transmit DF. By contrast, a study documented that wind velocity in the same month had a negative impact on the DF epidemic in Guangzhou [41]. Another study in Delhi, India identified an inverted J-shaped relationship between wind speed and DF, with the strongest effect appearing at 3 km/h [42]. Furthermore, the wind speed has not been significant correlated with dengue hemorrhagic fever in a recent Indonesian study [43]. This discrepancy may be explained by the difference in data types, modeling strategies, and geographical and demographical heterogeneity. The mechanism of the association between wind velocity and DF transmission warrants further investigation.

Our stratified analyses found that the impact of typhoons might vary by age-groups, such the elderly over 60 years old were more vulnerable than others. Current evidence indicates that the elderly is the most vulnerable group to most natural disasters, suffering from a disproportionate impact of tropical cyclones [44,45]. A study in New Jersey, the United States showed that older native residents might have the highest rate of house damage after Hurricane Sandy landed [46]. Without shelter, the elderly is more likely to suffer from adverse impacts of tropical cyclones on DF through increased contact with *Aedes* vectors. Moreover, several studies have reported the older age as a risk factor to be associated with dengue infection [47,48], which may be related to the impaired immune than the young adults. Hence, it is necessary to

pay more attention to elders in formulating tropical cyclone crisis management policy. Improving disaster response capacity and post-disaster resilience may reduce the risk of DF among the elderly.

We observed that the effect size of tropical cyclone on DF was highest in the first few days, followed by case deficits. This harvesting effect is not rare in environmental epidemiological studies [8,49]. For instance, a study in Guangdong Province exploring the association between tropical cyclones and childhood hand, foot and mouth disease had the similar finding [8]. One possible explanation is that the landing of tropical cyclones might cause rapid onset of DF among the most vulnerable population, which over-consumed the reservoir of susceptible individuals and decreased the risk of DF [50].

This study has several strengths. First, this is one of the few studies exploring the relationship between tropical cyclones and DF incidence worldwide. It is also the first study, to our knowledge, to explore the association in the PRD region, China. According to the DF surveillance data of NDSS, about 70% of DF cases in China occurred in the PRD region during the study period. This high proportion of DF incidence promises a reasonable representativeness of our study at the national level. Second, the use of reliable data source ensures the credibility of our research. The cases included in the study were all laboratory confirmed and clinical diagnostic cases, which reduced the bias of misdiagnosis. Although the incidence rate of DF might be underestimated due to the rigorous including criteria, this non-differential misclassification was unlikely to deviate the effect estimates substantially. Third, in contrast to the traditional case-control design, our use of time-stratified case-crossover method controlled for the temporal trend automatically [51]. In addition, we chose lag 5–14 days as the lag period based on the average incubation period of DF infection, rather than the goodness of fit of the model, making the result interpretation more biologically plausible. Finally, our results by intensity of tropical cyclones, sex and age-groups add to scientific evidence for better development of public health prevention and control strategies.

There are several limitations warrant to mention. In view of the nature of ecological time series research, the results of this study cannot be used as a strong causal evidence of tropical cyclone and DF incidence. As a preliminary study, we were unable to consider the impact of different landing sites and tracks of tropical cyclones on DF incidence in different cities, which needs exploration in the future. Furthermore, different wind direction of tropical cyclones may also cause various spatial patterns of local climate and mosquito distribution, thus potentially affecting the local transmission of DF. However, we were unable to adjust for the potential effect of wind direction in the model due to the lack of data. This issue warrants further exploration when data will be available in the future. In addition, only meteorological factors were included into the model due to the unavailability of mosquito-related data, the overlook of other important variables (e.g., vector density, host-species interactions) may influence the mechanism explanation of the impact of tropical cyclones on DF incidence. Tropical cyclone is a complex weather system that may increase DF transmission through one aspect of the system while mitigating transmission simultaneously through another [52]. Thus, further studies in finer geographical scale and by considering the complex dynamics at the climate-mosquito-virus interface are expected.

## Conclusions

In conclusion, our study demonstrates that tropical cyclones are associated with increased risk of indigenous DF incidence at lag 5–9 days in the PRD region with the elderly over 60 years most vulnerable. The findings highlight the importance of public health surveillance, preparedness, and response to dengue outbreaks during the tropical cyclone period. Mosquito-

control activities should be optimized in the aftermath of tropical cyclones to prevent the DF epidemic. Moreover, elderly should be given more consideration during tropical cyclones.

## Supporting information

**S1 Fig. Daily precipitation and number of DF cases in study periods of each tropical cyclone in the PRD from 2013 to 2018.**
(PDF)

**S2 Fig. Sensitivity analysis of the association between tropical cyclones and DF incidence on lag 5–14 days by using principal component analysis (PCA).**
(TIF)

**S3 Fig. Sensitivity analysis of the association between tropical cyclones and DF incidence on lag 5–14 days by changing meteorological variables.** Model I: original model; Model II: excluding RF; Model III: excluding EWV.
(TIF)

**S1 Table. Spearman correlation coefficients of tropical cyclone intensities and meteorological variables.**
(XLSX)

**S2 Table. Components extraction and total variance explained.**
(XLSX)

## Acknowledgments

We deeply appreciated the China Center for Disease control and Prevention and the National Meteorological Information Centre of China for sharing the data needed for the study.

## Author Contributions

**Conceptualization:** Chuanxi Li, Qi Zhao.

**Data curation:** Chuanxi Li, Qiyong Liu.

**Formal analysis:** Chuanxi Li.

**Funding acquisition:** Qiyong Liu, Wei Ma.

**Methodology:** Chuanxi Li, Qi Zhao, Zhe Zhao.

**Software:** Chuanxi Li, Qi Zhao.

**Supervision:** Qiyong Liu, Wei Ma.

**Validation:** Qiyong Liu, Wei Ma.

**Visualization:** Chuanxi Li, Qi Zhao, Wei Ma.

**Writing – original draft:** Chuanxi Li, Zhe Zhao.

**Writing – review & editing:** Qi Zhao, Qiyong Liu, Wei Ma.

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
