## [Decision Letter · Decision Letter 0]

22 Jul 2021

Dear Dr. Ma,

Thank you very much for submitting your manuscript "The association between tropical cyclones and dengue fever in the Pearl River Delta, China: A time-stratified case-crossover study" for consideration at PLOS Neglected Tropical Diseases. As with all papers reviewed by the journal, your manuscript was reviewed by members of the editorial board and by several independent reviewers. In light of the reviews (below this email), we would like to invite the resubmission of a significantly-revised version that takes into account the reviewers' comments. 

We cannot make any decision about publication until we have seen the revised manuscript and your response to the reviewers' comments. Your revised manuscript is also likely to be sent to reviewers for further evaluation.

Sincerely,

Alberto Novaes Ramos Jr

Associate Editor

Victor S Santos

Deputy Editor

Reviewer's Responses to Questions

**Key Review Criteria Required for Acceptance?**

**Methods**

-Are the objectives of the study clearly articulated with a clear testable hypothesis stated?

-Is the study design appropriate to address the stated objectives?

-Is the population clearly described and appropriate for the hypothesis being tested?

-Is the sample size sufficient to ensure adequate power to address the hypothesis being tested?

-Were correct statistical analysis used to support conclusions?

-Are there concerns about ethical or regulatory requirements being met?

Reviewer #1: Yes, the study clearly articulated.

I only asked to specify ICD codes for the health problems in the study.

Reviewer #2: (No Response)

Reviewer #3: (No Response)

Reviewer #4: -Are the objectives of the study clearly articulated with a clear testable hypothesis stated? Yes

-Is the study design appropriate to address the stated objectives? Yes

-Is the population clearly described and appropriate for the hypothesis being tested? Yes

-Is the sample size sufficient to ensure adequate power to address the hypothesis being tested? N/A

-Were correct statistical analysis used to support conclusions? Yes

-Are there concerns about ethical or regulatory requirements being met? Yes

**Results**

-Does the analysis presented match the analysis plan?

-Are the results clearly and completely presented?

-Are the figures (Tables, Images) of sufficient quality for clarity?

Reviewer #1: Yes, the applied methodology is correct.

Reviewer #2: (No Response)

Reviewer #3: (No Response)

Reviewer #4: -Does the analysis presented match the analysis plan? Yes

-Are the results clearly and completely presented? Yes

-Are the figures (Tables, Images) of sufficient quality for clarity? The resolution (dpi) of figures needs improvement.

**Conclusions**

-Are the conclusions supported by the data presented?

-Are the limitations of analysis clearly described?

-Do the authors discuss how these data can be helpful to advance our understanding of the topic under study?

-Is public health relevance addressed?

Reviewer #1: Yes, The results are reasonable and show strong association for lag 5.

Reviewer #2: (No Response)

Reviewer #3: (No Response)

Reviewer #4: -Are the conclusions supported by the data presented? Yes

-Are the limitations of analysis clearly described? Need improvement

-Do the authors discuss how these data can be helpful to advance our understanding of the topic under study? Yes

-Is public health relevance addressed? Yes

**Editorial and Data Presentation Modifications?**

Reviewer #1: As a reviewer I have the following remarks.

1. Abstract: line 23. I suggest to use the term “sex” rather than “gender”. 

2. Line 25: “and age-groups (<18y, 18-59y, =60y).” We are guessing that “y” is for years? In Line 194 is OK. 

3. Line 29: “relative risk (RR) of 1.62 (95%CI: 1.45-1.80)” – in general, we don’t introduce any abbreviations if the terms are used only one. Thus we don’t need RR and 95%CI, rather use a full spell of them. “relative risk of 1.62 with the 95% confidence interval (1.45-1.80)” or something a similar.

4. Line 72. Is possible to provide ICD-10 codes for these diseases?

5. Line 117. A trace of old reference style (24): “for 55.5% of the whole province’s population 24.”

6. Line 174. Yes, reference [27] is correct for the case-crossover technique, published in 1991. It took time (2005) to develop and justify time-stratified approach: Janes H, Sheppard L, Lumley T. Case-crossover analyses of air pollution exposure data: referent selection strategies and their implications for bias. Epidemiology. 2005 Nov;16(6):717-26. doi: 10.1097/01.ede.0000181315.18836.9d.

7. Table 2. “Sex”.

Thank you

Reviewer #2: (No Response)

Reviewer #3: (No Response)

Reviewer #4: (No Response)

**Summary and General Comments**

Reviewer #1: (No Response)

Reviewer #2: This study examined the association between tropical cyclones and dengue fever in the Pearl River Delta of China, using data from 2013-1018.The main conclusions are that, Tropical cyclones may increase the risk of local transmission of dengue fever in south

China, with the elderly more vulnerable. Generally, the paper is well written and all procedures and observations have been thoroughly explained in the discussion. There are some issues to be addressed:

1. Generalizability of this study and impact of the 2014 outbreak should be discussed.

2. There are six levels of tropical cyclone. In this study, tropical storm and severe tropical storm were combined into tropical storm category, typhoon, severe typhoon and super typhoon were classified into typhoon category. What happened to tropical depression?

3. Line 63: “Within in China” should be “In China”.

4. Line 64: “than any other locations” should be “than any other provinces”.

5. Line 84-85: the first sentence in this paragraph seems not logical here.

6. Line 172-173: “It eliminates the threat of control selection bias” better be replaced by “It reduces the selection bias”.

7. Line 280: “elderly was more affected than other age subgroups.” better be replaced by “elderly was more vulnerable than other age subgroups.”.

Reviewer #3: Dengue fever is of great public health concern in China affecting a large number of people every year. Although there are ample evidence of weather changes and dengue fever incidence, knowledge is lacking as to the impacts of weather-related disasters on dengue fever incidence. To address this important scientific topic, this study makes full use of recent dengue fever data in China to examine the association between tropical cyclones and dengue fever incidence. This manuscript is well written with important and interesting findings being clearly reported. I’ve few minor comments for authors to consider.

#1: Study period could be shown in the title.

#2: Abstract section: The aim of this study is unclear. I guess the authors sought to focus on dengue fever “incidence”. If so, this point needs to be clear.

#3: There are some grammar issues throughout this manuscript, which should be further polished up by a native English speaker.

#4: Lines 76-77: “As a typical imported disease…”, the meaning of this statement is unclear. Do you mean that dengue fever is a typical imported disease in China? Actually, most cases are local patients.

#5: Potential mechanism of tropical cyclones and dengue fever incidence could be briefly introduced in the Introduction section.

#6: More information is needed to clarify the selection of the studied months from June to October. Does the risky season mean the period with the highest incidence of dengue fever?

Reviewer #4: This study used time-stratified case-crossover study design to examine the association between tropical cyclones and dengue fever in the Pearl River Delta in China. In general, the article is logically and scientifically qualified with a certain innovation. Yet there are some issues need to be revised and addressed before accepted for publication.

1. In the Introduction (P5, Line 99), the authors mentioned using case-crossover design. As the methods part explained reasons of selecting the design, is it possible to add a few sentences describing using case-crossover design in previous studies (can be other vector borne diseases associated with cyclone and extreme weather) before express study aim? Thus the readers can better understand using the design even if there may be overlap with contents in Methods part. 

2. P7, 2.3 Disease data. The authors mentioned that the imported DF cases were excluded from this study. If the imported cases did not cause local transmission, this can be determined and sounds logically. The authors need to address it. 

3. P7, 2.4 Tropical cyclones and meteorological data. As the intensity (Beaufort level) of each cyclone may be changed (usually weakened after landing) during the process (e.g., a few days). As this study used maximum value by Beaufort level to describe intensity and determined time lag, is there some literature to support it?

4. P15, Line 274-276“This is one of the few studies exploring the relationship between tropical cyclones and DF worldwide. Meanwhile, this is the first study, to our knowledge, to explore the association in the PRD region, China.” This part can be relocated and integrated with strengths of this study. 

5. P282-283: “For instance, a study in Southeast China documented that tropical cyclone was likely to increase the risk of DF”. In this study, besides located in PRD, what other difference/new findings can be addressed compared with the SE China study?

6. P285, “Réunion”, better to be revised as “Réunion in the south of Indian Ocean”.

7. In describing limitations, the authors mentioned that the study “did not consider the impact (should also have “of” here) heterogeneity of different landing sites and different tracks of each tropical cyclone on the DF in different cities”. If considering spatial difference of impact of tropical cyclone on the DF, wind direction should also be considered as it may cause different spatial pattern of climate and mosquito distribution, thus influence spatiotemporal pattern of DF. Even in study considering PRD as a single site, wind direction may also potentially influence the pattern of DF (e.g., incidence). And this should also be addressed in the limitation. 

8. Some language errors/issues need to be corrected, e.g., P14, Line 258, “elderly” can be removed. The manuscript needs proofreading before being accepted. 

9. For figures, the resolution (dpi) should be increased.

PLOS authors have the option to publish the peer review history of their article (what does this mean?). If published, this will include your full peer review and any attached files.

Reviewer #1: No

Reviewer #2: No

Reviewer #3: No

Reviewer #4: No
---

## [Decision Letter · Decision Letter 1]

28 Aug 2021

Dear Dr. Ma,

We are pleased to inform you that your manuscript 'The association between tropical cyclones and dengue fever in the Pearl River Delta, China during 2013-2018: A time-stratified case-crossover study' has been provisionally accepted for publication in PLOS Neglected Tropical Diseases.

Best regards,

Alberto Novaes Ramos Jr

Associate Editor

Victor Santana Santos

Deputy Editor

Reviewer's Responses to Questions

**Key Review Criteria Required for Acceptance?**

**Methods**

-Are the objectives of the study clearly articulated with a clear testable hypothesis stated?

-Is the study design appropriate to address the stated objectives?

-Is the population clearly described and appropriate for the hypothesis being tested?

-Is the sample size sufficient to ensure adequate power to address the hypothesis being tested?

-Were correct statistical analysis used to support conclusions?

-Are there concerns about ethical or regulatory requirements being met?

Reviewer #1: Yes. The authors addressed raised issues concerning methodology.

In general, yes to all.

Reviewer #2: It is acceptable

Reviewer #3: (No Response)

Reviewer #4: Yes

**Results**

-Does the analysis presented match the analysis plan?

-Are the results clearly and completely presented?

-Are the figures (Tables, Images) of sufficient quality for clarity?

Reviewer #1: Yes - to all.

Reviewer #2: It is fine

Reviewer #3: (No Response)

Reviewer #4: Yes

**Conclusions**

-Are the conclusions supported by the data presented?

-Are the limitations of analysis clearly described?

-Do the authors discuss how these data can be helpful to advance our understanding of the topic under study?

-Is public health relevance addressed?

Reviewer #1: Yes- to all.

Reviewer #2: It is fine

Reviewer #3: (No Response)

Reviewer #4: Yes

**Editorial and Data Presentation Modifications?**

Reviewer #1: Accept

Reviewer #2: The authors have done a good job, suggest to accept it.

Reviewer #3: (No Response)

Reviewer #4: Accept

**Summary and General Comments**

Reviewer #1: The paper much improved after the first review.

The authors addressed the comments.

Reviewer #2: (No Response)

Reviewer #3: (No Response)

Reviewer #4: The authors have carefully replied the comments from the review and have revised the manuscript.

PLOS authors have the option to publish the peer review history of their article (what does this mean?). If published, this will include your full peer review and any attached files.

Reviewer #1: No

Reviewer #2: No

Reviewer #3: No

Reviewer #4: No

---

## [Editor Report · Acceptance letter]

6 Sep 2021

Dear Dr. Ma,

We are delighted to inform you that your manuscript, "The association between tropical cyclones and dengue fever in the Pearl River Delta, China during 2013-2018: A time-stratified case-crossover study," has been formally accepted for publication in PLOS Neglected Tropical Diseases.

Best regards,

Shaden Kamhawi

co-Editor-in-Chief

Paul Brindley

co-Editor-in-Chief
